# Rapid Detection of Anti-SARS-CoV-2 Antibodies with a Screen-Printed Electrode Modified with a Spike Glycoprotein Epitope

**DOI:** 10.3390/bios12050272

**Published:** 2022-04-26

**Authors:** Wilson A. Ameku, David W. Provance, Carlos M. Morel, Salvatore G. De-Simone

**Affiliations:** 1Oswaldo Cruz Foundation (FIOCRUZ), Center for Technological Development in Health (CDTS)/National Institute of Science and Technology for Innovation in Neglected Populations Diseases (INCT-IDPN), Rio de Janeiro 21040-900, RJ, Brazil; akira.ameku@gmail.com (W.A.A.); bill.provance@fiocruz.br (D.W.P.); carlos.morel@fiocruz.br (C.M.M.); 2Laboratory of Epidemiology and Molecualr Systematics (LESM), Oswaldo Cruz Institute, FIOCRUZ, Rio de Janeiro 21040-900, RJ, Brazil; 3Cellular and Molecular Department, Biology Institute, Federal Fluminense University, Niterói 24020-141, RJ, Brazil

**Keywords:** SARS-CoV-2, COVID-19, spike glycoprotein, epitope, electrochemical biosensor, point of care, immunological diagnostic

## Abstract

Background: The coronavirus disease of 2019 (COVID-19) is caused by an infection with severe acute respiratory syndrome coronavirus 2 (SARS-CoV-2). It was recognized in late 2019 and has since spread worldwide, leading to a pandemic with unprecedented health and financial consequences. There remains an enormous demand for new diagnostic methods that can deliver fast, low-cost, and easy-to-use confirmation of a SARS-CoV-2 infection. We have developed an affordable electrochemical biosensor for the rapid detection of serological immunoglobulin G (IgG) antibody in sera against the spike protein. Materials and Methods: A previously identified linear B-cell epitope (EP) specific to the SARS-CoV-2 spike glycoprotein and recognized by IgG in patient sera was selected for the target molecule. After synthesis, the EP was immobilized onto the surface of the working electrode of a commercially available screen-printed electrode (SPE). The capture of SARS-CoV-2-specific IgGs allowed the formation of an immunocomplex that was measured by square-wave voltammetry from its generation of hydroquinone (HQ). Results: An evaluation of the performance of the EP-based biosensor presented a selectivity and specificity for COVID-19 of 93% and 100%, respectively. No cross-reaction was observed to antibodies against other diseases that included Chagas disease, Chikungunya, Leishmaniosis, and Dengue. Differentiation of infected and non-infected individuals was possible even at a high dilution factor that decreased the required sample volumes to a few microliters. Conclusion: The final device proved suitable for diagnosing COVID-19 by assaying actual serum samples, and the results displayed good agreement with the molecular biology diagnoses. The flexibility to conjugate other EPs to SPEs suggests that this technology could be rapidly adapted to diagnose new variants of SARS-CoV-2 or other pathogens.

## 1. Introduction

Severe acute respiratory syndrome coronavirus 2 (SARS-CoV-2) has led to a global pandemic of coronavirus disease 2019 (COVID-19) [1]. Citizens of many countries were compelled to stay under partial or complete lockdown for months due to high transmissibility and disease severity [2]. Driven by the uncertainty concerning the effectiveness of rapidly deployed vaccines against severe disease, an absence of adequate therapies and appearance of new variants [1], diagnosis has played an important role in decision making. Case detection, monitoring, infection prevention, and supportive care are all tools for fighting the SARS-CoV-2 pandemic [1,3].

To keep up with our healthcare needs, it is crucial to develop a rapid point-of-care test to detect potential carriers of COVID-19 and those who have recovered as disease dynamics change at the speed of its spread [1]. While the gold standard technique for the diagnosis of the initial phase of an infection is the detection of viral genomic nucleic acid through reverse transcription-polymerase chain reactions (RT-PCR) from nasal and mouth swabs, its implementation in resource-limited settings is restricted due to infrastructure, skilled personnel, and time restrictions [1]. Serological assays for the presence of COVID-19-related immunoglobulin M (IgM) or IgG antibodies have provided a cost-effective and accurate method of tracking virus transmission to implement socio-political strategies against the spread of the contagion [1,4,5].

As a platform, electrochemical biosensors stand out to meet the demands for serological diagnostics through their characteristics: rapid, simple, portable, sensitive, easy-to-use, miniaturized, and compatible with portable instruments [6,7,8,9]. They can be combined with a wide range of biological recognition elements to detect clinically relevant compounds such as enzymes, nucleic acid, antibodies, epitopes, and others [10,11,12,13,14,15]. Among these choices, epitopes (EPs) hold a high interest since they represent the minimum amino acid sequences in a pathogen’s proteome that are bound by antibodies generated in a patient in response to an infection [7]. Their use can improve selectivity by the elimination of cross-reactivity based on sequence similarity to other pathogens, which is a major concern when whole antigens are used due to the presence of non-specific epitopes that can react with antibodies [16].

Here, the sensitivity of electrochemical measurements was combined with the specificity of EPs to develop an affordable biosensor for a serological assay to detect the presence of anti-SARS-CoV-2 antibodies as a diagnosis of COVID-19. A commercially available screen-printed electrode (SPE) was employed, while an epitope in SARS-CoV-2 spike (S) glycoprotein was employed as a binding target for antibody capture. The high performance in identifying infected patients and the absence of cross-reactivity suggests that his platform could be a viable solution for screening a large number of people. The flexibility of the proposed technology also presents the advantage of being rapidly adaptable to var-iants along with wide range of diseases by altering only the binding target.

## 2. Materials and Methods

### 2.1. Patient Samples and Project Approval

Positive controls consisted of fourteen serum samples from patients confirmed with COVID-19 by RT-PCR tests on nasopharyngeal or oropharyngeal swabs. Negative controls consisted of serum samples from blood bank donors (HEMORIO, Rio de Janeiro) collected before the pandemic (pre-November, 2019). For cross-reactivity, serum samples from patients diagnosed with Chagas disease, Dengue, Leshmaniose, and Chikungunya were used. For performance evaluations, a panel of 14 sera was used, collected from individuals who had suspected contact with individuals with COVID-19 and who were subsequently diagnosed by RT-PCR on nasopharyngeal or oropharyngeal swabs. Patient privacy was preserved by disassociating identifying information from the samples. The study was conducted following the International Coordinating Council for Clinical Trials and the Helsinki Declaration and was approved by the Local Ethics Committee UNIGRANRIO (No. 21362220.1.0000.5283) and Estacio de Sá University (No. 33090820. 8.0000.5284).

### 2.2. Chemicals and Reagents

Reagent-grade chemicals and peptide synthesis reagents were purchased from Sigma-Merck (St. Louis, MO, USA). Secondary antibodies were purchased from ThermoFisher (São Paulo, SP, Brazil). Hydroquinone diphosphate (diPho-HQ) salt was purchased from Metrohm Dropsens (Astúrias, Astúrias, Spain). All solutions were prepared with deionized water (>18.1 MΩ cm) obtained from a Nanopure Diamond system (Barnstead, Dubuque, IA, USA).

### 2.3. Solid-Phase Peptide Synthesis

Epitopes in the SARS-CoV-2 spike protein were prepared as amidated peptides on a Multipep-1 automated synthesizer (CEM Corp., Charlotte, NC, USA), as previously described [17]. Peptides were synthesized in a solid state on sintered glass filters containing Rink amide AM resin using the 9-fluorenylmethoxy carbonyl (F-moc) strategy. Finally, resin-bound peptides were deprotected and cleaved using trifluoroacetic acid and triisopropylsilane precipitated with diethyl ether and then lyophilized. Stock solutions were prepared with PBS and their concentration was determined by optical density and the theoretical molar extinction coefficient generated by the PROTPARAM software package (http://www.expasy.ch; accessed on 21 October 2021). Peptide sequences were confirmed by matrix-assisted laser desorption ionization time-of-flight mass spectrometry (MALDI-TOF MS).

### 2.4. Modification of the SPE’s Working Electrode

Carbon SPE working electrodes (DRP-110, Metrohm DropSend, Oviedo, Spain) were sensitized with peptides by a drop-casting method. First, the SPE was electrochemically treated in 0.1 mol L^−1^ of phosphate buffer solution (PBS), pH 7.4, applying +2 V (vs. Ag) for 60 s using a CompactState portable potentiostat (Ivium Technologies B.V., Eindhoven, Netherlands), rinsed with PBS, and allowed to dry at room temperature. Next, 2 µL of EP (100 µg mL^−1^ in PBS) was placed onto the surface of the SPE, followed by 10 µL of 2.5% (*w/w*) glutaraldehyde. After 30 min at room temperature, the peptide-modified SPEs were blocked overnight at 4 °C with 1% (*w/w*) BSA prepared in 0.1 mol L^−1^ PBS (pH 7.4).

### 2.5. Electrochemical Assay to Detect Antibodies COVID-19 Antibody IgG

The detection of anti-SARS-CoV-2 IgG antibodies was based on an indirect immunoassay wherein the subsequent binding of anti-human IgG conjugated with alkaline phosphatase (AP) hydrolyzed diPho-HQ to hydroquinone (HQ) that could be measured as electrochemical signals, shown schematically in Figure 1. Briefly, EP-specific IgGs (COVID-19 antibodies) present in 4 µL of diluted patient serum (1:100 in PBS with 1% BSA) were captured onto the sensitized working electrode surface of the SPE after a minimum 10 min incubation at 37 °C for complete drying. Then, SPEs were rinsed in PBS, and 4 µL of anti-human IgG secondary antibody solution conjugated with AP was added for another incubation to dry at 37 °C before rinsing in reaction buffer (0.1 mol L^−1^ Tris-HCl and 20 mmol L^−1^ MgCl_2_, pH 9.8). Next, the reaction buffer with 5 mM diPho-HQ was placed onto the SPE. After brief incubation (2 min), the presence of HQ was measured by square-wave voltammetry (SWV) with the following parameters: amplitude, 10 mV; frequency, 6.3 Hz; step, 10 mV; applied potential window, −0.5 to 0.2 V vs. Ag. Each cycle required 11 s, and a stable measurement was observed after the twentieth cycle (3.7 min total time).

### 2.6. Analysis of Blood Serum Samples

Electrochemical signals from positive and negative serum control samples were recorded to determine the cut-off value from Equation (1) (described in item 3.3) that was used to normalize all data as a reactivity index. A gray zone was defined as 1.0 ± 0.1. Next, sera were analyzed from persons with suspected contact with individuals diagnosed with COVID-19. Patient serum dilutions of 1:100 with PBS were used to perform the indirect immunoassays. The optimal dilution of 1:50,000 was employed for the secondary. Both incubations with antibodies were 8 min at 37 °C, which was sufficient to allow full evaporation of the applied solution. Differentially elevated currents were associated with serum from persons infected with SARS-CoV-2.

### 2.7. Statistical Analysis

One-way ANOVA tests were performed to evaluate the variation between prepared SPEs. Two-tailed Student’s t-tests with a confidence level of 95% were performed for pairwise comparisons [7,17,18].

## 3. Results

### 3.1. Development of Electrochemical Immunosensor

Previously, IgG linear B-cell epitopes (EPs) in the spike protein of SARS-CoV-2 were mapped by spot synthesis analysis [16]. Four of these were chosen to serve as antibody capture molecules on the surface of screen-printed electrodes. The intention was to develop an immunosensor utilizing a drop-casting approach with glutaraldehyde (GA) to sensitize the electrode surface. GA has been widely used to modify electrode surfaces due to its introduction of aldehyde functional groups that allow the covalent bonding of compounds containing terminal amino moieties such as EPs [19]. Single-use SPEs were fabricated by mixing PBS solutions containing a prospective EP and GA onto the surface of the electrode. After drying and washing, its ability to differentially detect anti-SARS-CoV-2 antibodies as a diagnosis for COVID-19 was evaluated. The formation of immunocomplexes with AP-conjugated sec-IgG antibodies allowed enzymatic reduction of diPho-HQ to generate HQ, a redox molecule measurable by square-wave voltammetry (SWV) [7].

Initially, the performance of SPEs conjugated to peptides EP1 (GPLQSYGFQPTG), EP2 (LPPLLTDEMIAQYTS), EP3 (GLDSKVGGNYNYG), and EP4 (RSYTPGDSSSGWTAG), which represent different EPs in the spike protein, was evaluated. The peak currents were recorded from measurements of an SWV while exposed to diluted serum samples from patients who tested positive and negative for COVID-19 (Figure 2A). From the ratio of the positive to negative peak currents (Figure 2B), the most robust measurement was obtained with EP2 as it demonstrated a significantly higher positive/negative signal ratio (*p*-value < 0.001) than the others. Therefore, EP2 was chosen for additional optimization.

Next, the production of the SPE was optimized. Figure 2B shows that the EP concentration on the SPE surface affected the current measured from the immunoreaction (Figure 2B). The ratio between SWV signals obtained after incubation in positive and negative samples significantly rose as increasing concentrations of EP were used (*p* < 0.01, Figure 2D), reaching a plateau at 100 µg mL^−1^ (*p* = 0.08). This suggested that antibodies were increasingly captured until reaching electrode surface saturation. Therefore, the most suitable EP concentration was 100 µg mL^−1^ and was used to produce all subsequent SPEs.

### 3.2. Optimization of Experimental Parameters, Reproducibility, and Stability

To optimize the analytical signal, the level of dilution for patient serum and secondary antibodies was evaluated. A fixed time of 16 min for the incubation times was chosen for antibodies. As the dilution of the positive serum sample was increased, there was a decrease in the signal from the positive samples (Figure 3A). Similarly, the non-specific binding of antibodies to the surface of the EP-sensitized SPE, represented by the negative controls, showed decreasing signals with higher dilutions. A maximum difference in the ratio of the SWV measurements obtained from positive and negative control sera was observed for sample dilutions of 1:100 (*p* < 0.001, Figure 3B), which was subsequently chosen as the optimal dilution factor. Differential signals between positive and negative samples were detectable up to a dilution factor of 1000 (*p* < 0.02), which suggested that the dynamic range of the approach was greater than 10-fold and could permit the detection of antibodies of low titers.

Another critical factor for detecting EP/IgG immunocomplexes was the concentration of the secondary antibodies. Small decreases in the positive signals over the range of secondary antibody concentrations suggested that its presence is not a limiting factor to the measurement (Figure 3C). However, a large difference in the measurement of negative sera suggests there is a potential for non-specific background signals at higher concentrations. The background signal significantly decreased to the lowest levels at a dilution of 1:50,000 (*p* < 0.02), which did not meaningfully impact the signal from the positive control (comparison between 1:5000 and 1:30,000 and between 1:30,000 and 1:50,000 provided *p* = 0.08 and 0.30, respectively) and provided the most prominent signal-to-noise ratio (Figure 3D).

### 3.3. Biosensor Performance

The reproducibility of the SPEs was analyzed by evaluating electrodes prepared on different days using the same protocol. A relative standard deviation (RSD) of 5% in the SWV HQ response was calculated from three measurements of a 1:100 diluted positive serum on the electrodes prepared on different days, which demonstrated the practical reproducibility of the method (Figure 4B). SPEs prepared on the same day were stored at 4 °C in PBS to test stability. After 14 days of storage, the signal obtained from a positive sample diluted at 1:100 showed an RSD of 7% (*n* = 3) and decay in the average response of 10% compared to using an SPE prepared on the same day. The measurements showed that the response was preserved statistically (*p* = 0.1) (Figure 4B). However, the SWV current decreased by 20% and presented an RSD of 8% (*n* = 3) after 30 days of storage. These levels suggested that the performance of the SPE was significantly decreased compared to the same day as prepared (*p* = 0.01). Thus, these biosensors had only a 14-day shelf life at 4 °C.

To evaluate the performance of the EP2-sensitized SPEs, a confirmed panel of positive (*n* = 14) and negative (*n* = 17) patient samples (1:100 in PBS) previously assayed by a commercial assay for COVID-19 (DiaPro Diagnostic Bioprobes Srl, Mi, Italy) was measured (Figure 4B). Using Equation (1), a cut-off was calculated as 1.4 µA for detecting COVID-19.
(1)Cut-off=a·X+f·SD 
where *X* is the mean and SD is the standard deviation of independent negative control readings, and *a* and *f* are two arbitrary multipliers, which were 1 and 3 in the present case, respectively [20,21]. The final results were normalized as the ratio of the sampled signal to the cut-off value (S/co). A gray zone was defined between ±10% of 1.0 wherein results between 0.9–1.1 were considered indeterminate, >1.1 as positive, and <0.9 as negative. Cross-reactivity was evaluated with sera collected from patients diagnosed with Chagas disease, dengue, leshmaniose, and chikungunya. Each presented a S/CO value of less than 0.9 which corresponds to a specificity of 100%. Positive controls collected from those confirmed with COVID-19 analyzed previously by RT-PCR demonstrated a median S/CO ratio near 1.7 (Figure 4B, dark blue). To simulate a real-world application for the diagnosis of COVID-19, 14 samples from persons with suspected contacts were assayed with the SPE. From these, 13 volunteers were considered positive and 1 was in the gray zone for selectivity of 93%.

## 4. Discussion

Serological diagnostic tests that utilize the presence of antibodies as the definition of whether an individual was infected by a pathogen detect antibodies in their serum or blood. This can be conducted as a lab-based assay such as an enzyme-linked immunosorbent assay (ELISA) [7] or chemiluminescent immunoassay [22], which are time-consuming and expensive, or a point-of-care test based on lateral flow technology that can show limited sensitivity [23,24]. We propose an electrochemical assay based on commercially available SPEs that can be easily sensitized to capture diagnostic antibodies. We began with the diagnosis of COVID-19 by choosing to sensitize the SPE with a peptide that represents an epitope in the spike protein to capture anti-SARS-CoV-2 IgG antibodies. The S protein is a highly exposed part of the viral structure [3] and contains several immunodominant epitopes [1,23]. The use of a single epitope allows for the development of highly specific serological assays. Furthermore, a focus on IgG antibodies, which are more prevalent than IgM antibodies [1,3], enabled improvements in sensitivity and specificity [1]. Ultimately, sensitized SPEs can be easily fabricated at a low cost that is compatible with portable equipment to provide rapid results in non-laboratory settings.

Generally, an electrochemical immunosensor consists of an electron-conducting solid surface where the molecule of interest (antibody or antigen) can be selectively captured, and its presence detected by the amplification [7] or suppression [25] of an electrochemical signal. A key element of the biosensor is the biological component that can be immobilized onto its surface, such as an antigen [25] or EP [7]. Here, the SPE was modified with a peptide representing an EP specific to the S protein of SARS-CoV-2. No cross-reactivity was displayed against serum from patients with Chagas, chikungunya, leishmaniosis, and dengue (Figure 3B), all pathogens endemic to Rio de Janeiro, Brazil [26]. The single sample that presented a S/CO ratio in the gray zone was from an individual whose blood sample was collected soon after presenting symptoms that may not have been seroconverted. Overall, within the electrochemical platform, the biosensor showed high specificity and sensitivity of 100% and 93%, respectively.

The requirement of a small patient sample combined with a low cost could make it economically viable to perform multiple assays to ensure a confident result [1,3]. The diagnostic accuracy of different assays is variable [27,28,29,30] and a recent work compared the performance of ten assays. Sensitivities of 40–77% (65–81% considering IgG plus IgM) were found [31], making our electrochemical platform a competitive methodology for the diagnosis of COVID-19. In addition, the proposed device provided a reliable method for detecting COVID-19 IgG in serum and was rapid (22 min compared to >90 min required for an ELISA), even though the time spent to modify the substrates with antigens and block them was comparable in both mentioned techniques. Notably, the measurements can be performed in volumes lesser than 100 µL, the required volume for an ELISA assay, which translates to 2 nL of serum that can be easily acquired from a single finger prick sample of blood. Furthermore, the simplicity of electrode preparation, ease of use, accuracy, and low cost all suggest that this platform could be utilized as a point-of-care diagnostic assay to detect infected individuals for observation and to track the spread of disease, as well as identify high titer samples for recruiting potential donors to provide convalescent plasma therapy [32,33]. Considering that the relevance of the information obtained from the measurements was defined by the peptide used to sensitize the SPE, this platform has a high potential to be modified towards other pathogens and aspects of the COVID-19 pandemic, such as antibody titer post-vaccination, temporal changes in antibody response, and altered reactivity to variants.

## 5. Conclusions

We developed a portable and affordable biosensor to rapidly detect COVID-19 infection by detecting anti-SARS-CoV-2-specific IgG antibodies in patient serum. Based on electrochemical reactivity, it showed a sensitivity of 93% and a specificity of 100%. Single-use electrodes were fabricated by the surface modification of commercially available SPE with a peptide that represents an epitope-specific SARS-CoV-2 spike glycoprotein. The immobilized peptide could capture COVID-19-specific IgG antibodies for measurement by an indirect immunoassay using an enzyme-conjugated secondary IgG that hydrolyzed diPho-HQ into HQ, a redox molecule detectable by SWV. Under optimized conditions, it differentiates infected and non-infected individuals that correlated with RT-PCR diagnosis. No cross-reactivity was displayed to other pathogens such as *Trypanosoma cruzi* (Chagas disease), chikungunya, leishmaniosis, and dengue. The biosensor platform has the flexibility to meet the demands of other pathogens and their respective diseases

## 6. Patents

The peptide described in this study is protected under Brazilian and US provisional patent applications BR 10.2019.017792.6 and PCT/BR2020/050341, respectively, filed by FIOCRUZ, and may serve as a future source of funding.

## Figures and Tables

**Figure 1 biosensors-12-00272-f001:**
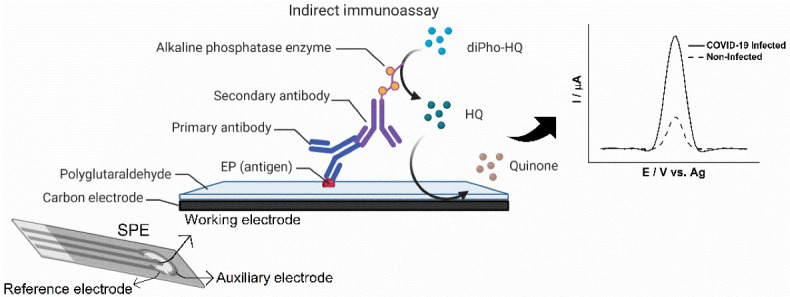
Schematic representation of a positive indirect immunoassay to detect COVID-19. Anti-SARS-CoV-2 IgG antibody (primary antibody) in patient serum samples is captured onto the surface of the SPE’s peptide-modified working electrode. Retained human antibodies are bound by AP-conjugated anti-human antibodies (Secondary antibodies). Enzymatic activity converts diPho-HQ into HQ, which can be measured by square-wave voltammetry.

**Figure 2 biosensors-12-00272-f002:**
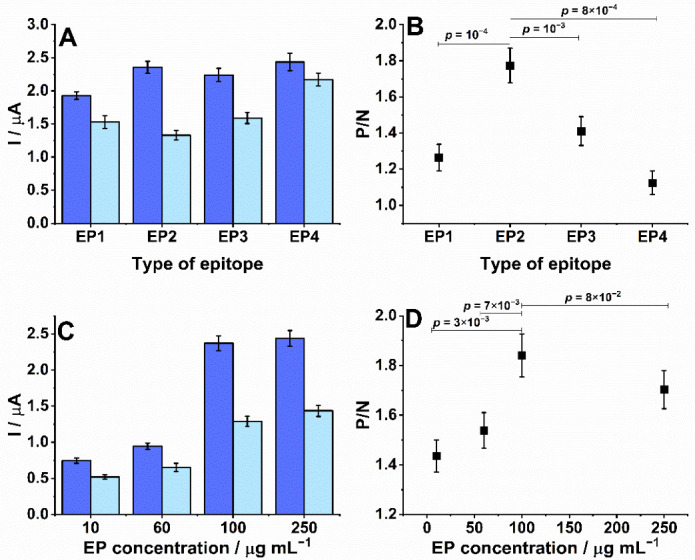
Target choice for conjugation to SPE and measurement optimization. SPEs were sensitized with peptides that represented four EPs identified in the SARS-CoV-2 spike protein. Next, SPEs were incubated with patient sera diluted 1:100 in PBS for 10 min at 37 °C before rinsing. SPEs were similarly incubated with an AP-labeled anti-human secondary (1:30,000), rinsed and presented with 5 mM diPho-HQ in 100 mM Tris-HCl and 20 mM MgCl2 (pH 9.8) for its enzymatic conversion to HQ. (**A**) Peak currents measured by SWV from positive (blue) or negative (light blue) sera using SPE sensitized with EP1-EP4. (**B**) Ratio between positive and negative signals (P/N) from graph A. (**C**) Peak currents measured from positive (blue) and negative (light blue) sera with SPEs sensitized over a range of EP2 peptide concentrations (10–250 μg mL^−1^). (**D**) Ratio between positive and nega-tive signals (P/N) for data in graph C. All experiments were performed in duplicate. Solution vol-umes were 2 μL for antibody solutions and 50 μL for washes. Antibody incubations were for 8 min at 37 °C. The parameters of SWVs for amplitude, frequency, step, and applied potential window were 10 mV, 6.3 Hz, 10 mV, −0.6–0.6 V (vs. Ag), respectively.

**Figure 3 biosensors-12-00272-f003:**
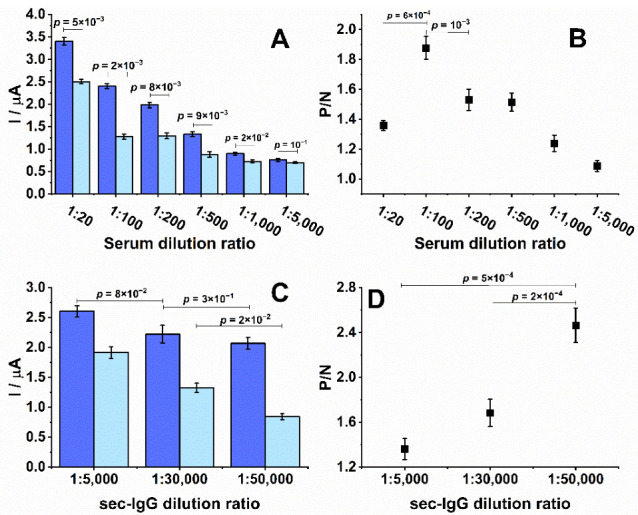
Optimization of primary and secondary antibody dilutions. All SPEs were sensitized with 100 μg/mL of EP2. (**A**) Peak currents measured over a dilution range of positive (blue) and negative (light blue) patient sera. (**B**) Ratio between positive and negative signals (P/N) from graph A. (**C**) Peak currents measured from positive (blue) and negative (light blue) patient sera diluted 1:100 in combination with a range of dilutions of the secondary antibody. (**D**) Ratio between positive and negative signals (P/N) from graph C. All experiments were performed in duplicate.

**Figure 4 biosensors-12-00272-f004:**
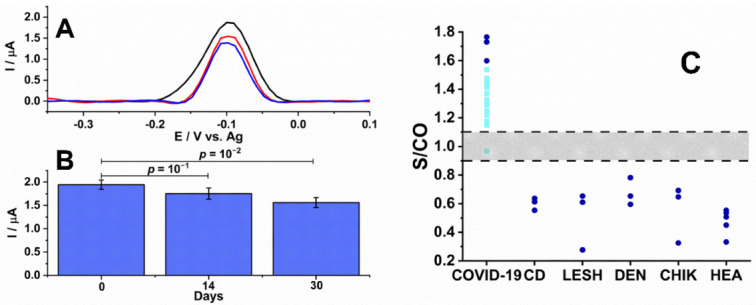
Stability, performance, and cross-reactivity of EP2 conjugated SPEs. (**A**) SPE reproducibility was evaluated by fabricating multiple SPEs according to the optimized protocol and performing SWVs in positive serum (1:100) on the day of fabrication (black line, *n* = 3), after 14 days of storage at 4 °C (red line, *n* = 3) and 30 days of storage at 4 °C (blue line, *n* = 3). (**B**) The graph shows the variations in recordings. (**C**) For real-world applications, the cut-off was determined using positive and negative controls (dark blue). Multiple serum samples from persons suspected of having COVID-19 (light blue); individuals with Chagas disease (CD; *Trypanosoma cruzi*), leishmaniosis (LESH), dengue (DEN), and chikungunya (CHIK); and healthy (HEA) patients were assayed. The gray region represents ±10%.

## Data Availability

The data presented in this study are available upon request from the corresponding author.

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
