# Peer review of "Rapid Detection of Anti-SARS-CoV-2 Antibodies with a Screen-Printed Electrode Modified with a Spike Glycoprotein Epitope"

_biosensors, 2022, doi:10.3390/bios12050272_

Round 1

Reviewer 1 Report

In this study, the authors have developed an electrochemical biosensor for the rapid detection of serological immunoglobulin (Ig) G antibody in sera against the Spike protein that is based on the B-cell epitope (EP) specific to SARS-CoV-2 spike glycoprotein. The assay was structured as an indirect ELISA that measures the product hydroquinone. The biosensor's performance evaluation indicated high selectivity and specificity for COVID-19 with no cross-reactivity against Chagas disease, Chikungunya, Leishmaniosis, and Dengue. This technology could become a promising tool to diagnose new variants of SARS-CoV-2 or other pathogens. However, I can see a major issue in the experimental structure, no standard curve with SARS-CoV-2 commercially available antibodies has been developed. Thus, the biosensor's LODs and LOQs could not be presented. 

In addition, I also have two other minor comments:

- Please explain better the procedure in the sentence (lines 135-136):

"Briefly, EP-specific 135 IgG in 4 μl of dilute patient serum (1:100 in PBS with 1% BSA) was captured onto the sensitized working electrode surface of the SPE."

- Line 274: N=? How many positive samples

Author Response

Reviewer #1 (changes highlighted in light blue in the revised text)

 1. Please explain better the procedure in the sentence (lines 135-136):

"Briefly, EP-specific 135 IgG in 4 μl of dilute patient serum (1:100 in PBS with 1% BSA) was captured onto the sensitized working electrode surface of the SPE." 

R: We are grateful for the valuable suggestion. We rewrote the mentioned sentence.

2. Line 274: N=? How many positive samples

R: We appreciated the accurate observation. The N value is 3, as pointed out in the text.

Reviewer 2 Report

To deal with the current healthcare crisis resulted from the coronavirus disease (COVID-19), rapid diagnostic methods for detecting SARA-COV-2 virus are highly needed. In this manuscript, Ameku et al., developed an electrochemical biosensor for analyzing serological immunoglobulin (Ig) G antibody captured using linear B-cell epitope (EP) specifically against the Spike protein of SARA-COV-2 virus. The resultant performance of their EP-based biosensor demonstrated a selectivity of 93% and a specificity of 100% for SARA-COV-2. Moreover, the cross-reactivity against other diseases such as Chagas disease, Chikungunya, Leishmaniosis, and Dengue was not observed. In conclusion, this manuscript provides a low-cost platform for SARA-COV-2 diagnosis using human sera. With the following issues being addressed, the Reviewer would suggest its publication in Biosensors.

  1. The reported method is a qualitative measurement of Ig G antibodies against SARA-COV-2 present in human serum. Compared to conventional ELISA assay, it does not shorten the assay time when taking into consideration the time spent to prepare peptide-functionalized SPE. The author should acknowledge the limitation in Discussion. The comparison of EP-based biosensor to other reported methods should also be included.
  2. The concentration of stock solution can be quantitatively confirmed with conventional ELISA assay. Line 109 “Stock solutions were prepared with PBS and their concentration determined by optical density”.
  3. It is not clear why the authors allowed the SPE’s working electrode to dry at room temperature after surface modification with EPs or incubation with serum or antibodies? Will this cause denature of the antibodies?
  4. It will be clearer to add an indicator for the bars in Figure 2 and Figure 3. Significance comparison should also be given in the figures.
  5. The actual image of the whole setup of EP-based biosensor should be given.
  6. There are some typos, such as Line 45, “of vaccines aga inst severe disease and transmission to new variants along with an”, Line 68, “which a major concern when whole antigens are used due to the presence of non-specific”, Line 77, “Dengue suggests that his platform could be a viable solution to screening large number of”, and Line 121, “The solution was allowed it dry at room temperature and after 30 min presented a gel-like appearance”.

Author Response

1.The reported method is a qualitative measurement of IgG antibodies against SARA-COV-2 present in human serum. Compared to conventional ELISA assay, it does not shorten the assay time when taking into consideration the time spent to prepare peptide-functionalized SPE. The author should acknowledge the limitation in the Discussion. The comparison of EP-based biosensors to other reported methods should also be included.

R: We appreciated the observations. The time spent to functionalize SPE was comparable to conventional ELISA to guarantee the efficiency in the blocking step and, thus, avoid the non-specific binding. However, the time spent during the sample assays for electrochemical measures (22 min.) was significantly lesser than ELISA assay (90 min.). We considered the mentioned limitation in the text.

Regarding the raised question about the comparison, we were not able to develop the analytical curve and, thus, determine the limit of detection and quantification that hampered the comparison to other reported methods. It was not allowed since we selected a linear epitope of Spike protein that interacts specifically with a sort of COVID-19 IgG employing spot synthesis analysis using RT-PCR-confirmed positive and negative serum samples. It means that the commercially available antibodies do not interact with the selected epitope, or the specificity/selectivity is compromised hindering the determination of those analytical figures of merit. In this way, for this purpose, we are purifying and quantifying the captured antibodies for a future report. For now, in this work, we present a qualitative sensor for COVID-19 diagnosis given the emergency of the pandemic. 

  1. The concentration of the stock solution can be quantitatively confirmed with a conventional ELISA assay. Line 109 “Stock solutions were prepared with PBS and their concentration determined by optical density”.

R: We would like to thank the advice. We will arrange for the incorporation of the ELISA assay in our EP quantification protocols. Today, we quantify the EP amount through optical density since it provides fast results and is compatible with our laboratory routine.

  1. It is not clear why the authors allowed the SPE’s working electrode to dry at room temperature after surface modification with EPs or incubation with serum or antibodies? Will this cause denature of the antibodies?

R: We appreciated the question. We allowed the SPE to dry at room temperature for convenience, the attached small droplets from the rinse solution evaporated relatively fast without needing a higher temperature. It was preferable to work with dried electrodes so as not to interfere with the solution concentration used in the subsequent step.    

  1. It will be clearer to add an indicator for the bars in Figure 2 and Figure 3. Significance comparison should also be given in the figures.

R: We are grateful for the valuable suggestion. The meaning of the bar's colors was mentioned in the figure titles. The p values between comparisons were added in Figures 2 and 3.  

  1. The actual image of the whole setup of the EP-based biosensor should be given.

R: We appreciated the suggestion. The method’s setup was added in Figure 1. 

  1. There are some typos, such as Line 45, “of vaccines against severe disease and transmission to new variants along with an”, Line 68, “which is a major concern when whole antigens are used due to the presence of non-specific”, Line 77, “Dengue suggests that his platform could be a viable solution to screening a large number of”, and Line 121, “The R: the solution was allowed it dry at room temperature and after 30 min presented a gel-like appearance”

R: We are grateful for the valuable observations. We corrected the text.  
